# Association of impaired kidney function with mortality in rural Uganda: results of a general population cohort study

Robert Kalyesubula [1,2,3] Isaac Sekitoleko,[3] Keith Tomlin,[1]
Christian Holm Hansen,[4] Billy Ssebunya,[3] Ronald Makanga,[3]
Moses Kwizera Mbonye,[3] Janet Seeley [5] Liam Smeeth,[1] Robert Newton [3,6]
Laurie A Tomlinson [1]

For numbered affiliations see end of article.

**Correspondence to**
Robert Kalyesubula;
rkalyesubula@gmail.com

## ABSTRACT

**Objective** To determine the association between baseline kidney function and subsequent all-cause mortality.

**Design and setting** A general population-based cohort study from rural Uganda.

**Participants** People aged 18 years and above with measured baseline estimated glomerular filtration rate (eGFR), recruited from survey rounds in 2011–2012 or 2014–2015 and followed up to March 2019.

**Outcome measure** The primary outcome was all-cause mortality, identified through reports from community health workers and verified by verbal autopsy. The association between baseline eGFR category and mortality was determined using multivariable Cox regression.

**Results** Of 5812 participants in both rounds, we included 5678 (97.7%) participants with kidney function and mortality data; the median age was 36 years (IQR 24–50), 60.7% were female, 10.3% were hypertensive, 9.8% were HIV-positive and 1.5% were diabetic. During a median follow-up of 5.0 years (IQR 3.7–6.0) there were 140 deaths. In age-adjusted and sex-adjusted analyses, eGFR <45 mL/min/1.73 m$^2$ at baseline was associated with a 5.97 (95% CI 2.55 to 13.98) increased risk of mortality compared with those with baseline eGFR >90 mL/min/1.73 m$^2$. After inclusion of additional confounders (HIV, body mass index, diabetes, hypertension, alcohol and smoking status) into the model, eGFR <45 mL/min/1.73 m$^2$ at baseline remained strongly associated with mortality (HR 6.12, 95% CI 2.27 to 16.45), although the sample size fell to 3102. Test for trend showed strong evidence (p<0.001) that the rate of mortality increased progressively as the category of baseline kidney function decreased. When very high eGFR was included as a separate category in age-adjusted and sex-adjusted analyses, baseline eGFR ≥120 mL/min/1.73 m$^2$ was associated with increased risk of mortality (HR 2.68, 95% CI 1.47 to 4.87) compared with the reference category of 90–119 mL/min/1.73 m$^2$.

**Conclusion** In a prospective cohort in rural Uganda we found that impaired baseline kidney function was associated with subsequently increased total mortality. Improved understanding of the determinants of kidney disease and its progression is needed in order to inform interventions for prevention and treatment.

### Strengths and limitations of this study

► This is the first data of its kind from sub-Saharan Africa, generated in a large, well-established population cohort with robust standardised procedures for detailed measurements of covariates such as blood pressure and with creatinine measured according to recommended standards.

► There is high participation and retention in the study within the local community and regular reporting of mortality and migration by local study workers, leading to limited loss to follow-up.

► There were missing data for covariates such as smoking, diabetes and blood pressure for about one-third of the participants, so complete case analysis led to reduction in power for the fully adjusted model.

► Participants in the 2011–2012 round, who were more likely to have had complete data for potential confounders, also had longer follow-up time over which they could have died.

► We used baseline measures of covariates and were not able to update health-related confounders over time.

► There is likely to be inaccuracy in the measurement of kidney function, which we categorised using estimated glomerular filtration rate calculated from serum creatinine, a breakdown product of muscles and related to muscle mass and dietary meat intake.

## INTRODUCTION

Chronic kidney disease (CKD) affects approximately one in every ten adults across the world and is strongly associated with morbidity and mortality.[1 2] The leading causes of death among people with CKD include cardiovascular disease and infections, while a proportion progress to end-stage kidney disease (ESKD).[3 4] In high-income countries, ESKD is a chronic disease that can be managed with dialysis or kidney transplantation.[5] However, in low-income countries such as Uganda there is very limited access

## Summary box

### What is already known
► Impaired kidney function is independently associated with subsequent all-cause mortality in cohorts from high-income countries.
► It is not known if this was the case in sub-Saharan Africa, where there is a young population, a high burden of infectious disease mortality and low prevalence of many non-communicable diseases.

### What this study adds
► This study demonstrates that impaired baseline kidney function is associated with all-cause mortality in a general population cohort of adults in rural Uganda.
► This suggests that kidney function plays a key role in the overall health status in sub-Saharan Africa and should be considered a public health priority.

to kidney replacement therapies and most patients with ESKD die prematurely.[6 7] Compounding the challenge of managing kidney disease in sub-Saharan Africa (SSA) is that population prevalence estimates are limited,[1] biochemical methods of measuring kidney function have been suboptimal,[8] and the appropriate way to estimate kidney function is uncertain since equations developed in high-income countries have not been validated for SSA.[8–11] In order to determine the value of developing health services to measure and manage kidney disease, we need to understand the relative importance of kidney function in determining prognosis compared with other chronic diseases in this setting. At present no population-based studies have examined the association between kidney function and mortality in SSA. Therefore, we sought to determine the association between baseline kidney function and all-cause mortality in a large, well-described general population cohort (GPC) of people in rural Uganda from 2011 to 2019.

## METHODS
### Study design and setting
This was a prospective GPC of adults in Kyamulibwa, a rural community 132 km from Kampala, the capital of Uganda. The population consists of rural subsistence farmers, with a few periurban dwellers, and is similar to the broader rural populations of Uganda, which constitute about 76% of the country.[12] The GPC was established in 1989 by the UK Medical Research Council and the Uganda Virus Research Institute to study the epidemiology of HIV infection.[13] Subsequently, other diseases of interest, including non-communicable diseases, have been examined in biannual rounds in the same cohort. It is an open cohort with new births, migrations (inward and outward) and deaths recorded regularly by community volunteers through the annual census.

For this study we selected a subset of the GPC using information from the 2011–2012 and 2014–2015 rounds of the GPC in which a major disease of interest was CKD and thus baseline information on kidney function was available.[14] The outcome was total mortality among participants who had a baseline creatinine measured in either 2011–2012 or 2014–2015.

### Patient and public involvement in research
Research within the GPC is carried out with participation from members of the public of Kyamulibwa who participate in a community advisory board, which is involved from inception of studies to their implementation. The community is consulted on the priorities of research for the area, and through seminars and face-to-face meetings community members are involved in the development of key questions and outcome measures. In relation to kidney disease, we have developed simple tools on how the kidneys work and conducted awareness campaigns in the villages to explain and get input from the participants and other members of the community. We plan to disseminate the results of both of this work and those of other kidney-related projects with the community of Kyamulibwa.

### Participants
We identified potential participants aged above 18 years who were recruited consecutively from household visits in each village. Questionnaires were administered and samples collected at research hubs set up in homesteads of a community member in each location.

### Variables
Each round of the GPC records baseline demographic characteristics including age, sex, tribe and maximum level of education as well as HIV status using an approved national algorithm.[15] We used participants' height and weight to determine their body mass index (BMI) using weight $(kg)/height^2 (m^2)$. We tested for creatinine level using a Cobas e 601 auto analyser (Roche Diagnostics, North America), using the enzymatic method traceable to an isotope dilution mass spectrometry method.[14 16] We calculated estimated glomerular filtration rate (eGFR) based on the CKD-Epi creatinine equation without adjustment for ethnicity[17] and classified impaired kidney function in categories analogous to those used to define CKD stages.[18] Detailed descriptions of these measurements have been previously published.[14]

Further to these near complete characteristics, in keeping with the different disease areas investigated in different rounds of the GPC, we also collected information on marital status, smoking, alcohol use, diabetes and blood pressure. This information was the focus of the 2011–2012 round, so data collection was near complete but was also collected, although not completely, in 2014–2015. Blood pressure was measured using a digital sphygmomanometer (Omron M4-1) after the participant was seated and the mean of the second and third readings taken at 5 min intervals was used for analysis. Participants with systolic blood pressure ≥140 mm Hg and/or diastolic blood pressure ≥90 mm Hg and those who were on treatment for hypertension were classified as hypertensive.

Diabetes mellitus was defined as having Heamoglobin A1C (HbA1C) >6.5%, being previously diagnosed with diabetes or being on current treatment for diabetes.

We collected data on mortality through registers updated monthly through reports from community health workers, verified by verbal autopsy (a method used to ascertain the cause of a death based on an interview with next of kin or other caregivers used where no other routine systems are in place and where many people die at home), conducted in accordance with standard guidelines by the WHO and cross-checked via annual census.[19]

### Statistical analysis

We conducted a complete case analysis with regard to baseline creatinine measurements and also excluded participants with missing data regarding outcome or with probable linkage errors. We described population characteristics according to predefined categories of eGFR. We present the rate of mortality per 1000 person-years and use multivariable Cox modelling to determine the HR and 95% CI for mortality for each category of eGFR. In Cox regression we included participants up to their last point of follow-up, so participants who later migrated contributed follow-up time to the analysis while they were present in the area. Participants included in the baseline round also contributed follow-up time if they returned to the area after a period of outmigration.

We tested the proportional hazards assumption using log-minus-log survival plots and visual inspection of Kaplan-Meier curves. Due to incomplete data for some covariates, we present sequentially adjusted models to demonstrate the effect of additional covariate adjustment on the relationship between eGFR and mortality. Thus, we present models adjusted for age and sex, and then additionally for baseline HIV status, hypertension, diabetes mellitus, BMI, smoking, alcohol and marital status. These potential confounders were defined a priori based on existing literature showing their association with low eGFR and mortality in this population[14 20] and were included after investigation of collinearity and data sparsity.

Because HIV is common in this population and has been associated with increased risk of both kidney impairment and mortality, we examined potential interaction between baseline eGFR and HIV status in the association with mortality by fitting an interaction term in the final adjusted model.

In additional analysis, to examine for a 'J-shaped curve' in the association between eGFR and mortality, we repeated the main analysis after recategorisation of eGFR with those $\geq 120$ mL/min/1.73 m$^2$ as a separate group. We analysed all data using STATA V.15.0 statistical software.

All study participants gave written informed consent to participation, specimen storage and future use of their stored samples.

### RESULTS

Of 5812 participants in both rounds, we included 5678 (97.7%) with kidney function and mortality data available for the analysis (figure 1). The median age of the participants was 36 years (IQR 24–50 years), with most being female (60.7%). Among those with data available,

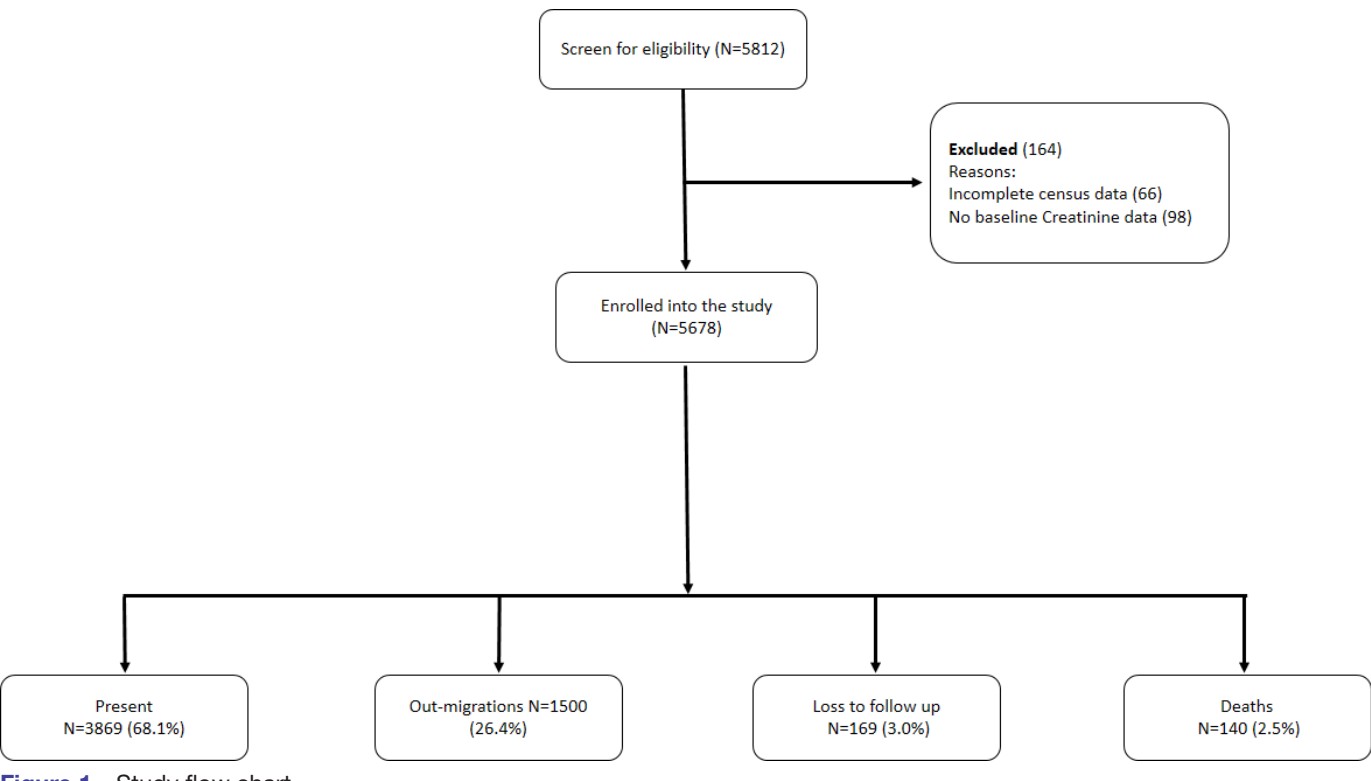

**Figure 1** Study flow chart.

70.2% were of normal weight, 9.8% were HIV-positive, 90.5% were non-smokers, 58.5% did not consume alcohol, 10.3% were hypertensive and 1.5% were diabetic (table 1). Comparison of baseline characteristics between participants eligible for inclusion from the 2011–2012 and the 2014–2015 rounds is shown in online supplemental table 1. We also compare the baseline characteristics of participants by five-level stages of kidney function, including a separate category for people with eGFR >120 mL/min/1.73 m$^2$, in online supplemental table 2.

During a median follow-up of 5.0 years (IQR 3.7–6.0) there were 140 deaths. The incidence rate for participants with eGFR >90 mL/min/1.73 m$^2$, eGFR 60–90 mL/min/1.73 m$^2$, eGFR 45–60 mL/min/1.73 m$^2$ and eGFR <45 mL/min/1.73 m$^2$ was 3, 9, 19 and 63 deaths per 1000 person years at risk (PYAR), respectively. During follow-up, 26.4% (1500 of 5678) of the participants migrated and 3.0% (169 of 5678) were lost to follow-up.

Age-adjusted, sex-adjusted and fully adjusted associations with mortality are shown in table 2 and figure 2. In age-adjusted and sex-adjusted analyses, eGFR <45 mL/min/1.73 m$^2$ at baseline was associated with a 5.97 (95% CI 2.55 to 13.98) increased risk of mortality compared with those with baseline eGFR >90 mL/min/1.73 m$^2$. After inclusion of additional confounders into the model, eGFR <45 mL/min/1.73 m$^2$ at baseline remained strongly associated with mortality (HR 6.12, 95% CI 2.27 to 16.45). The test for trend showed strong evidence (p<0.001) that the rate of mortality increased progressively as the category of baseline kidney function decreased.

There was no evidence of an interaction between HIV and kidney function with risk of mortality in the fully adjusted model (p=0.672).

In an additional analysis, when very high eGFR was included as a separate category, there was weak evidence of a 'reverse J-shaped curve' (online supplemental figure 1 and online supplemental table 3). In age-adjusted and sex-adjusted analyses, baseline eGFR ≥120 mL/min/1.73 m$^2$ was associated with increased risk of mortality (HR 2.68, 95% CI 1.47 to 4.87) compared with the reference category of 90–119 mL/min/1.73 m$^2$. However, in fully adjusted analyses, the CI crossed the null (HR 1.65, 95% CI 0.61 to 4.44).

## DISCUSSION

In a rural Ugandan population cohort, we found a graded association between low baseline kidney function and subsequent mortality. Participants with severe kidney impairment (eGFR <45 mL/min/1.73 m$^2$) had a more than sixfold risk of dying compared with those with eGFR >90 mL/min/1.73 m$^2$. This was despite a low prevalence of obesity, diabetes and regular smoking, all key risk factors for kidney disease in high-income countries.

Our study has a number of strengths. It is the first data of its kind from SSA, generated in a large, well-established population cohort with robust standardised procedures for detailed measurements of covariates such as blood

pressure and with creatinine measured according to recommended standards.[13] There is high participation and retention in the study within the local community and regular reporting of mortality and migration by local study workers, leading to limited loss to follow-up.[13]

However, there were also limitations. This cohort covers a region that is mostly rural and may not be generalisable to urban regions. However, our study population is in keeping with the majority of people living in Uganda and in many regions of SSA. There were missing data for covariates such as smoking, diabetes and blood pressure for about one-third of the participants, so complete case analysis led to reduction in power for the fully adjusted model. In addition, participants in the 2011–2012 round, who were more likely to have had complete data for potential confounders, also had longer follow-up time over which they could have died. However, marginal differences between the effect estimates for the association of eGFR categories with mortality between the age-adjusted, sex-adjusted and fully adjusted models suggest that neither confounding nor selection bias due to variable follow-up has substantially impacted on our findings of the association between baseline kidney function and mortality. We used baseline measures of covariates and were not able to update health-related confounders over time. There is likely to be inaccuracy in the measurement of kidney function, which we categorised using eGFR calculated from serum creatinine, a breakdown product of muscles and related to muscle mass and dietary meat intake.[16] Therefore, the prevalence of low BMI in this population where food scarcity is common[21] means that we may have underestimated the prevalence of impaired glomerular filtration rate (GFR) in this population. In addition, validation of GFR estimating equations is currently limited in SSA.[8] We measured kidney function only once while two measures 3 months apart are required to confirm a diagnosis of CKD, which may have led to misclassification of the level of kidney function for some participants.[18] In addition, we do not have measures of proteinuria, an important early marker of kidney damage, so we cannot comment on the association with mortality in this setting. Migration and subsequent loss to follow-up among younger, healthier participants may have led to selection bias with over-representation of older participants with health problems, including impaired kidney function, remaining in the cohort. However, people with chronic health problems often return home, so we anticipate capturing the majority of deaths among the baseline cohort, even among those who previously migrated. Finally, we do not know the causes of death and the small number of people in lower kidney function subgroups led to limited power and imprecise estimates with wide CIs.

Recent community-based studies of the associations of kidney disease suggest important differences in the prevalence and associations of impaired kidney function between countries in SSA. Those including participants from urban areas in South Africa and Nairobi found similar risk factors for those well established in

**Table 1** Baseline characteristics of participants categorised by level of kidney function in the general population cohort (N=5678)

| Variable | Estimated glomerular filtration rate (mL/min/1.73 m$^2$) | | | | |
| --- | --- | --- | --- | --- | --- |
| | ≥90, n (%) 4563 (80.4) | 60–89, n (%) 1022 (18.0) | 45–59, n (%) 73 (1.3) | <45, n (%) 20 (0.4) | Total, n (%) 5678 (100) |
| **Sex** | | | | | |
| Male | 1853 (40.6) | 343 (33.6) | 26 (35.6) | 7 (35.0) | 2229 (39.3) |
| Female | 2710 (59.4) | 679 (66.4) | 47 (64.4) | 13 (65.0) | 3449 (60.7) |
| Age (years), median (IQR) | 32 (22–44) | 56 (45–70) | 66 (58–76) | 68 (42–76) | 36 (24–50) |
| **Age groups** | | | | | |
| <35 | 2531 (55.5) | 76 (7.4) | 3 (4.1) | 3 (15.0) | 2613 (46.0) |
| 35–44 | 946 (20.7) | 176 (17.2) | 4 (5.5) | 2 (10.0) | 1128 (19.9) |
| 45–54 | 619 (13.6) | 210 (20.6) | 10 (13.7) | 3 (15.0) | 842 (14.8) |
| 55–64 | 311 (6.8) | 214 (20.9) | 14 (19.2) | 1 (5.0) | 540 (9.5) |
| 65–75 | 133 (2.9) | 186 (18.2) | 19 (26.0) | 5 (25.0) | 343 (6.0) |
| >75 | 23 (0.5) | 160 (15.7) | 23 (31.5) | 6 (30.0) | 212 (3.7) |
| **HIV status** | | | | | |
| Negative | 4134 (90.6) | 901 (88.2) | 70 (95.9) | 13 (65.0) | 5118 (90.1) |
| Positive | 424 (9.3) | 117 (11.5) | 3 (4.1) | 7 (35.0) | 551 (9.7) |
| Missing | 5 (0.1) | 4 (0.4) | 0 (0.0) | 0 (0.0) | 9 (0.2) |
| **BMI classification\* (kg/m$^2$)** | | | | | |
| Underweight (<18.5) | 504 (11.1) | 143 (14.0) | 14 (19.2) | 4 (10.0) | 665 (11.7) |
| Normal weight (18.5–24.99) | 3201 (70.2) | 623 (61.0) | 41 (56.2) | 14 (70.0) | 3879 (68.3) |
| Overweight (25.0–29.99) | 552 (12.1) | 166 (16.2) | 14 (19.2) | 2 (10.0) | 734 (12.9) |
| Obese (>30.0) | 167 (3.7) | 77 (7.5) | 3 (4.1) | 0 (0.0) | 247 (4.4) |
| Missing | 139 (3.1) | 13 (1.3) | 1 (1.4) | 0 (0.0) | 153 (2.7) |
| **Diabetes mellitus†** | | | | | |
| No | 2998 (65.7) | 815 (79.8) | 62 (84.9) | 16 (100.0) | 3891 (68.5) |
| Yes | 62 (1.4) | 20 (2.0) | 3 (4.1) | 0 (0.0) | 85 (1.5) |
| Missing | 1503 (32.9) | 187 (18.3) | 8 (11.0) | 4 (20.0) | 1702 (30.0) |
| **Hypertension‡** | | | | | |
| Normal | 2756 (60.4) | 614 (60.1) | 35 (48.0) | 9 (45.0) | 3414 (60.1) |
| Hypertensive | 319 (7.0) | 228 (22.3) | 30 (41.1) | 7 (35.0) | 584 (10.3) |
| Missing | 1488 (32.6) | 180 (17.6) | 8 (11.0) | 4 (20.0) | 1680 (29.6) |
| **Smoking status** | | | | | |
| Not current smokers | 2801 (61.4) | 748 (73.2) | 55 (75.3) | 13 (65.0) | 3617 (63.7) |
| Non-daily smokers | 64 (1.4) | 30 (2.9) | 1 (1.4) | 1 (5.0) | 96 (1.7) |
| Daily smokers | 209 (4.6) | 64 (6.3) | 9 (12.3) | 2 (10.0) | 284 (5.0) |
| Missing | 1489 (32.6) | 180 (17.6) | 8 (11.0) | 4 (20.0) | 1681 (29.6) |
| **Alcohol consumption** | | | | | |
| Infrequent/no alcohol | 1911 (41.9) | 393 (38.5) | 29 (39.7) | 4 (20.0) | 2337 (41.2) |
| Regular alcohol | 1164 (25.5) | 449 (43.9) | 36 (49.3) | 12 (60.0) | 1661 (29.3) |
| Missing | 1488 (32.6) | 180 (17.6) | 8 (11.0) | 4 (20.0) | 1680 (29.6) |
| **Marital status** | | | | | |
| Single/widowed | 853 (18.7) | 444 (43.4) | 37 (50.7) | 13 (65.0) | 1347 (23.7) |
| Married | 2496 (54.7) | 536 (52.5) | 33 (45.2) | 7 (35.0) | 3072 (54.1) |
| Missing | 1214 (26.6) | 42 (4.1) | 3 (4.1) | 0 (0.0) | 1259 (22.2) |

\*Body mass index (BMI) classification according to the WHO (weight/height$^2$; kg/m$^2$).
†Diabetes mellitus defined as heamoglobin A1C (HbA1C) >6.5%, previously diagnosed with diabetes mellitus or being on treatment for diabetes.
‡Hypertension was defined as having diastolic blood pressure ≥90 mm Hg, systolic blood pressure ≥140 mm Hg or being on treatment for hypertension.

**Table 2** Results of age-adjusted, sex-adjusted and fully adjusted regression models for the association between kidney function and mortality in the general population cohort

| | Age-adjusted and sex-adjusted | | Fully adjusted* | |
| --- | --- | --- | --- | --- |
| | HR (95% CI)<br>n=5678 | P value | HR (95% CI)<br>n=3102 | P value |
| eGFR (mL/min/1.73 m$^2$) | | | | |
| ≥90 | Reference | | Reference | |
| 60–89 | 1.11 (0.73 to 1.71) | | 1.21 (0.73 to 2.03) | |
| 45–59 | 1.72 (0.81 to 3.69) | | 1.91 (0.80 to 4.54) | |
| <45 | 5.97 (2.55 to 13.98) | <0.001 | 6.12 (2.27 to 16.45) | 0.003 |
| Age (years) | 1.06 (1.05 to 1.07) | <0.001 | 1.05 (1.03 to 1.07) | <0.001 |
| Sex | | | | |
| Male | Reference | | Reference | |
| Female | 0.50 (0.36 to 0.70) | <0.001 | 0.70 (0.42 to 1.07) | 0.188 |
| HIV status | | | | |
| Positive | | | 1.67 (0.87 to 3.18) | 0.120 |
| Negative | | | Reference | |
| Hypertension | | | | |
| Normotensive | | | Reference | |
| Hypertensive | | | 1.08 (0.58 to 2.01) | 0.433 |
| Diabetes mellitus | | | | |
| No | | | Reference | |
| Yes | | | 1.11 (0.35 to 3.57) | 0.861 |
| BMI classification | | | | |
| Underweight | | | 1.71 (1.10 to 2.67) | |
| Normal weight | | | Reference | |
| Overweight | | | 0.20 (0.05 to 0.84) | |
| Obese | | | 0.35 (0.48 to 2.55) | 0.006 |
| Marital status | | | | |
| Currently married | | | 0.77 (0.48 to 1.23) | 0.273 |
| Single/widowed | | | Reference | |
| Alcohol consumption | | | | |
| Regular alcohol | | | 1.50 (0.92 to 2.46) | 0.104 |
| No/infrequent alcohol | | | Reference | |
| Smoking status | | | | |
| Not current smokers | | | Reference | |
| Non-daily smokers | | | 1.22 (0.50 to 2.97) | |
| Daily smokers | | | 1.63 (0.93 to 2.87) | 0.236 |

*Fully adjusted HR adjusted for age, sex, HIV status, hypertension, diabetes mellitus, BMI, marital status, alcohol use and smoking.
BMI, body mass index; eGFR, estimated glomerular filtration rate.

high-income countries: age, hypertension, diabetes, HIV and female sex.[22] However, in cohorts in Malawi and Uganda, where the populations are younger, often living in rural areas and with low levels of smoking, obesity and diabetes showed low prevalence of impaired kidney function and an association with diabetes was found, possibly due to hyperfiltration.[14 23] These findings suggest that the aetiology of kidney disease in rural SSA is unclear and

highlight the possibility that the mechanism by which kidney function is associated with increased mortality may differ from that in high-income countries.

Prospective studies that have looked at the association between mortality and kidney disease in SSA are limited to cohorts which are hospital-based,[24–27] included patients with only ESKD[28] or HIV,[29] and studied albuminuria alone,[30] and many are also limited to short-term

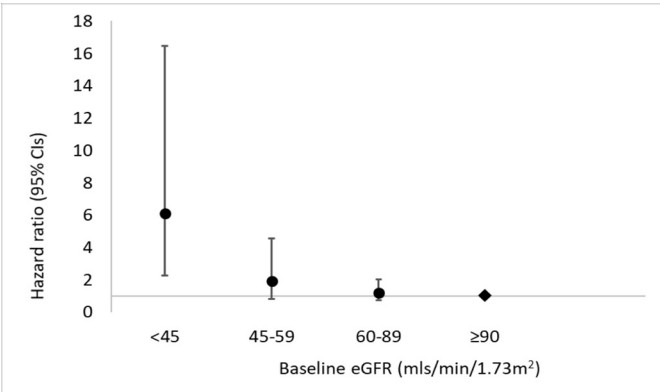

**Figure 2** HR and 95% CI for the fully adjusted associations of baseline estimated glomerular filtration rate (eGFR) and mortality in a rural Ugandan population cohort.

follow-up. Our study is consistent with results from high-income countries which have shown a consistent, strong and graded association between impaired kidney function and risk of subsequent mortality.[31] These studies also demonstrate an increase in mortality in people at the highest level of eGFR, in a reverse 'J-shaped curve'. Our findings are again consistent, demonstrating weak evidence of an increase in mortality in people with the highest category of baseline eGFR. Further confirmation will be needed in larger cohorts with longer follow-up.

While HIV can cause kidney disease,[32] we did not find an interaction between HIV, impaired kidney function and mortality, suggesting that low kidney function has the same pattern of association with mortality among people who are HIV-negative. Unexpectedly, we did not find an association between hypertension and mortality in our study despite evidence that it is a key driver of disease and outcomes in SSA and globally.[33] It is possible that repeated screening within this cohort population has led to higher rates of treatment and control than in other populations, and the prevalence of hypertension in this region (10.3%) was lower than the estimated national prevalence of 26.4%.[34] However, it is also possible that the causal relationship between hypertension and kidney damage in SSA has been oversimplified due to lack of kidney function measurement and longitudinal follow-up. Recent discovery of strong genetic risks for kidney disease in African-Americans has suggested that hypertension and kidney dysfunction may develop in part from shared risk factors.[35] Similarly, diabetes is an independent predictor of mortality among patients with kidney disease and is the leading cause of CKD globally.[36] However, in this population with a low prevalence of diabetes, it was not associated with mortality.

In conclusion, in this prospective cohort study based in a rural Ugandan community, we found that baseline impaired kidney function was associated with mortality, which remained after adjusting for known shared risk factors such as diabetes, hypertension and tobacco and despite the coexistent burden of high infectious disease mortality. This suggests that kidney function plays a key role in overall health status in SSA and, since options for treatment of kidney failure are very restricted, should be included within public health targets. Improved understanding of the determinants of kidney disease and its progression is needed in order to inform interventions for prevention and treatment.

**Author affiliations**
[1]Epidemiology and Population Health, London School of Hygiene & Tropical Medicine, London, UK
[2]Physiology and Internal Medicine, Makerere University College of Health Sciences, Kampala, Uganda
[3]Non-Communicable Disease Epidemiology, MRC/UVRI and LSHTM Research Unit, Entebbe, Uganda
[4]Infectious Diseases Epidemiology, London School of Hygiene & Tropical Medicine, London, UK
[5]Department of Global Health and Development, London School of Hygiene & Tropical Medicine, London, UK
[6]Department of Health Sciences, University of York, York, UK

**Acknowledgements** We thank Rose Nabwato (study nurse), Josephine Kagina Nabukenya (study nurse), Vincent Arumadri (laboratory technician), Francis Namugera (data clerk), Eva Nambi Ssejjemba (study administrator) and Grace Godfrey Sseremba (community mobiliser) for the role they played in the study.

**Collaborators** African Research on Kidney Disease (ARK) Study: ARK Uganda: Dr Robert Kalyesubula (PI), Professor Robert Newton, Professor Janet Seeley, Professor Anatoli Kamali, Dr Gershim Asiki, Professor Pontiano Kalebi, Professor Moffat Nyirenda, Dr Sylvia Kushemererwa, Dr Billy Ssebunya, Ronald Makanga, Dominic Bukenya and Isaac Sekitoleko; ARK Malawi: Professor Amelia C Crampin Dr Wisdom Nakanga, Dr Jospehine Prynn, Dr Joseph Mkandawire and Louis Banda; ARK South Africa: Professor Stephen Tollman, Professor Saraladevi Naicker (co-PI South Africa), Professor Jaya Anna George, Professor Michele Ramsay, Dr Alisha Wade, Professor Jonathan Levin, Professor Kathleen Kahn, Professor Shane Norris, Dr June Fabian (co-PI South Africa), Sanushka Naidoo, Cassandra Soo, Tracy Snyman and Lungile Khambule. Collaborating centre: UK LSHTM, Professor Liam Smeeth, Dr Laurie Tomlinson and Dr Dorothea Nitsch.

**Contributors** RK conceived the idea. RK, KT, JS, RN and CHH advised on the methodology. JS, RN, LS and LAT obtained funding. RK, KT, BS, RM, MKM, JS and RN collected and curated the data and administered the project. RK and IS analysed the data. RK and LAT wrote the original draft and all authors contributed to and approved the final manuscript. RK takes full responsibility for the work and the conduct of the study, had access to the data, and controlled the decision to publish. RK is the guarantor of this article.

**Funding** This work was funded by GlaxoSmithKline Africa Non-Communicable Disease Open Lab (grant number: 8111) as part of a broader multicentre collaborative study between South Africa, Uganda, Malawi and the London School of Hygiene & Tropical Medicine, which is collectively identified as the African Research on Kidney Disease (ARK) Network. This work was also funded by Wellcome Trust Intermediate Clinical Fellowship (grant number 101143/Z/13/Z) awarded to LAT. The general population cohort is funded by the UK Medical Research Council (MRC) and the UK Department for International Development (DFID) under the MRC/DFID Concordat agreement and is also part of the EDCTP2 programme supported by the European Union. This work was also funded by THRiVE-2, DELTAS Africa (grant number DEL-15-011) from Wellcome Trust (grant number 107742/Z/15/Z) and the UK government, which supported JS. The funders had no role in study design, data collection and analysis, decision to publish, or preparation of the manuscript.

**Competing interests** None declared.

**Patient and public involvement** Patients and/or the public were involved in the design, or conduct, or reporting, or dissemination plans of this research. Refer to the Methods section for further details.

**Patient consent for publication** Not required.

**Ethics approval** This study involves human participants and was approved by the Uganda Virus Research Institute (UVRI) Research and Ethics Committee (UVRI-REC-#HS 1978), the Uganda National Council for Science and Technology (UNCST-#SS 4283), and the London School of Hygiene & Tropical Medicine Observational/Interventions Research Ethics Committee (LSHTM Ethics #21802). Participants gave informed consent to participate in the study before taking part.

**Provenance and peer review** Not commissioned; externally peer reviewed.

**Data availability statement** Data are available upon reasonable request. Owing to data protection concerns, there are restrictions on access to the underlying data. The GPC database contains 25 years of longitudinal data sets on demographics and disease surveillance. All data (census, survey and laboratory) generated through the cohort are stored and curated at the MRC/UVRI and the LSHTM Research Unit. Data access for specific research purposes is possible and has been granted previously. For any data access enquiries, you may contact the director, MRC/UVRI and the LSHTM Research Unit or by email to mrc@mrcuganda.org or the corresponding author.

**ORCID iDs**
Robert Kalyesubula http://orcid.org/0000-0003-3211-163X
Janet Seeley http://orcid.org/0000-0002-0583-5272
Robert Newton http://orcid.org/0000-0001-6715-9153
Laurie A Tomlinson http://orcid.org/0000-0001-8848-9493

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
