## [Reviewer comments · BMJ Open]

ARTICLE DETAILS

TITLE (PROVISIONAL)	Association of impaired kidney function with mortality in rural Uganda: results of a general population cohort study
AUTHORS	Kalyesubula, Robert; Sekitoleko, Isaac; Tomlin, Keith; Hansen, Christian; Ssebunya, Billy; Makanga, Ronald; Mbonye, Moses Kwizera; Seeley, Janet; Smeeth, Liam; Newton, Robert; Tomlinson, Laurie

VERSION 1 – REVIEW

REVIEWER	Mayr, Michael University Hospital Basel, Medical Outpatient Clinic
REVIEW RETURNED	02-Jun-2021

GENERAL COMMENTS	The authors performed a prospective general population based cohort study with the aim to analyse the association between baseline kidney function and mortality in a rural region of Uganda. This topic is very important as epidemiological and outcome data are still scarce and important for public health strategies. However, there are relevant points that require careful revision. Major points Could you describe the recruitment process in more detail? How was it done? Was it random? How many participants refused to participate? In which context and for what purpose were blood samples and clinical parameters taken? Further, the measurements of the lab values should be described briefly. How was the blood pressure measured? The authors used multivariable Cox modelling to determine the Hazard Ratio (HR) for mortality for each category of eGFR. It would be of interest to continuously analyze the association between eGFR and mortality. In some covariates, a relevant number of data are missing (marital status, alcohol consumption, smoking status, socioeconomic status, hypertension, and diabetes mellitus). The authors should address how many participants were included in the regression models (age and sex, and fully adjusted), see results, table 2 and abstract. These data are important numbers. The number of 5678 participants (see abstract) might misleading regarding the main purpose of the study. Additionally, these data should also be shown in the study flow, i.e. how many participants were available for the regression models. Further, how was the follow-up on the people who migrated? Were these parts of the analyses? If the authors make a statement such as “Comparison of characteristics with included participants showed that those with missing data were younger but otherwise there were no important
--

differences (Supplementary Table 1)", then p-values should be shown.

Patients characteristics from the participants analyzed in regression model should be displayed as table 1. In a supplementary table, these data can be compared with people who could not be taken into account due to missing data (with p-values). This table would replace supplementary table 2.

The quality of the figures is rather low. Please revise them. The manuscript should be carefully revised in term of wording and logic within sentences.

Discussion

"However, there are no substantial differences in characteristics between those with complete and incomplete covariate information". I think, the authors did not compare these two populations. They compared those with complete and those with all complete/non complete information (Supp Table 1).

Others

It seems, trial registration number is missing.

Minor points

Use always the term kidney instead renal (kidney function, kidney impairment and so)

Abstract

Paragraph "participants" should be more precise (recruitment between 2011 and 2014), compare background: "We collected data on subsequent mortality for participants who had a baseline creatinine measured in either 2011-2012 or 2014-2015."

Please, specify: Overall? Median follow-up (IQ and range): "We registered 140 deaths with a median follow-up of 5.0 years"

Please, add the HR for the subgroup of eGFR ≥ 90 mls/min/1.73m²: "Adjusting for age and sex, HIV, hypertension, diabetes, BMI, marital status, and alcohol and tobacco use participants with eGFR ≤ 45 mls/min/1.73m² had six-fold higher mortality compared to those with eGFR ≥ 90 mls/min/1.73m² (HR 6.12 (95% CI 2.27-16.45))."

"In a prospective cohort with high rates of follow-up we found that baseline kidney function was associated with subsequently increased mortality in a graded manner." Please revise: baseline kidney function was associated with mortality (because baseline kidney function is not per se associated with increased mortality).

Strengths and limitations of this study

Revise this paragraph. The phrases should be very clear and precise.

Please revise the following paragraph (wording) and please note, in this study, kidney function was not a covariate, but rather the main variable: This is a large well-established population cohort with robust standardised procedures for detailed measurements of covariates such as kidney function and blood pressure, and creatinine was measured according to recommended international standards.

Background

"Chronic kidney disease (CKD) affects approximately one in every ten adults in high-income countries and is strongly associated with morbidity and mortality 1 2." Is the study by Stanifer et al the right citation for "high- income countries"?

	“In high-income countries, ESKD is a chronic disease that can be managed with dialysis or kidney transplantation” Is the study by Bukabau et al the right quote for “high-income countries”? Patient and Public Involvement in Research “We also undertook a qualitative study to understand the way community members appreciated kidney disease and this work has been published elsewhere”, the part “and this work has been published elsewhere” may be deleted. Participants “who had been recruited from household visits to a homestead in the community”, please revise the wording. Variables “and classified impaired kidney function in categories analogous to those used to define CKD stages” please cite the more recent guidelines from 2012. Statistics Could you specify what you mean, could you give an example? “Confounders were decided a priori based on research regarding associations of chronic conditions with eGFR in this population” Typing error “s”? 120mls/min/1.73m² Results The incidence rate of death for the participants enrolled in the study was 4 deaths per 1000 person-years at risk (PYAR). Please, specify “overall incidence rate”. For participants with eGFR >90mls/min/1.73m² the incident rate was about 3 deaths per 1000 (PYAR) person-years at risk and for those <60mls/min/1.73m² it was 27 deaths per 1000 PYAR. Please avoid imprecise wording in the result section, e.g. “about”. Discussion “which may have led to misclassification of some participants 17.” Replace the reference with the guidelines 2012/13. “obesity and diabetes showed low prevalence of impaired renal function and found a protective association with diabetes, possibly due to hyperfiltration.” In this context it seems better not to mention the term “protective”.
--	--

REVIEWER	Muiru, Anthony University of California San Francisco
REVIEW RETURNED	06-Jul-2021

GENERAL COMMENTS	Multiple studies from the Global North have demonstrated that individuals with Chronic kidney disease (CKD) are at high risk for cardiovascular and all-cause mortality. Even though the burden of CKD in the Global South is rising, few studies have evaluated the long-term outcomes associated with CKD, and and none in sub-Saharan Africa. Therefore, the work by Kalyesubula et al, is timely and fills a critical gap in the literature. Strengths of this study include the use of a well-established large cohort (over 5,000 participants), long-term follow up (with low rates of lost to follow up [other than out migration]), a rural population, and use of standardized kidney function measurements. The use of IDMS calibrated creatinine measurements and the omission of Black race coefficient in CKD-EPI equation to estimate GFR are important
--

	additional strengths. The main outcome was ascertained using a validated approach and seems appropriate. The major weakness of this work is the inability to ascertain causes of death, and lack of albuminuria and follow up kidney function measurements. I have a few minor questions.  1. Why was eGFR of 45 used as the cutoff point for comparison? why not 60 or 30? 2. Risk factors: It would be useful to include more details about risk factors control if possible, for example, are markers of HIV control available? Are measures of hypertension control available, this would help strengthen the authors hypothesis "It is possible that repeated screening within this cohort population has led to higher rates of treatment and control than in other populations". 3. Out migration: The authors have paid a great deal of attention to the selection bias introduced by the high rate of migration. As they point out, there are important differences between individuals who migrated out of these rural communities and those who remained including age. Is it possible to obtain data from these individuals who have migrated from study communities, for example determine their vital status? 4. On table 1: It would informative to include the total N(%) by eGFR stage 5. It is notable that higher eGFR (>120) was associated with mortality. One possible theory is hyperfiltration (as one would see in patients with sickle cell disease (SCD), though life expectancy for SCD patients is quite low in Uganda and wouldn't expect SCD patients to be included in this study or early diabetes but again the prevalence here is much lower). Another possibility is that low creatinine (i.e. higher eGFR) is indicative of undernutrition, hence the higher mortality. The authors can consider non-creatinine markers of kidney disease such as cystatin C for future exploration. 6. The HR associated with being overweight is notable for being "protective" as is the HR for obese although the CI crosses the null. Would postulate why overweight appears to be protective.
--	--

VERSION 1 – AUTHOR RESPONSE

Reviewer: 1

Dr. Michael Mayr, University Hospital Basel

Comments to the Author:

The authors performed a prospective general population based cohort study with the aim to analyse the association between baseline kidney function and mortality in a rural region of Uganda. This topic is very important as epidemiological and outcome data are still scarce and important for public health strategies. However, there are relevant points that require careful revision.

Thank you for the review. We have made extensive changes in all the recommended sections as detailed responses to each of the issues raised. We have revised the abstract to include the numbers of members included in the final model. We have revised the methods section to include more details on the recruitment process, laboratory and clinical measurements.

We have revised the results section to include analysis on the participants who were included in the final model and also revised the tables to reflect these changes.

We have included a new supplementary table for the extra analysis requested to explain the direct relationship between eGFR and mortality.

Our discussion section has been updated to explain the changes and concerns raised by the reviewers.

Major points

Could you describe the recruitment process in more detail? How was it done? Was it random? How many participants refused to participate? In which context and for what purpose were blood samples and clinical parameters taken? Further, the measurements of the lab values should be described briefly. How was the blood pressure measured?

We have included more details on the recruitment process and on how the measurements and tests were conducted as requested by the reviewer. We used consecutive sampling for all eligible participants and research hubs are set up in homesteads of community members where the questionnaires are administered and the samples are collected. In 2011-12 and 2014-15, the major foci were diabetes, hypertension, obesity and chronic kidney disease. These diseases were selected because they are among the common non-communicable diseases that have been found to cause high morbidity and mortality. Our participant response rate at baseline was high, at 97.2%, we included 5678 participants out of the available 5842 participants enrolled from census conducted in 2011-12 and 2014-15. Please see methods section **pages 5 and 6**.

The authors used multivariable Cox modelling to determine the Hazard Ratio (HR) for mortality for each category of eGFR. It would be of interest to continuously analyze the association between eGFR and mortality.

Thank you for this input. We used Martingale residuals to confirm that quadratic functional form is better than the linear as shown in the curves and table below. The smoothed line appears approximately flat at zero in model 2. However, the quadratic functional form of the eGFR provides better model fit than the linear. We have included this extra analysis as part of the supplementary appendix and added a statement in the results to this effect. " On analysis for the association between kidney function as continuous eGFR and mortality, there was very strong evidence ($P < 0.001$) of a non-linear association between eGFR and mortality largely re-enforcing the U-shaped relationship (**Supplementary Table 3**). Please see **page 7** in the analysis section of the methods section and the results section on **page 11**.

Supplementary Table 3. Results of age-sex and fully-adjusted regression models for the association between kidney function (continuous eGFR) and mortality in the general population cohort.

	Number of participants	Age and sex adjusted		Fully adjusted [#] (n=3102)	
		HR (95% CI)	P-value	HR (95% CI)	P-value
eGFR (mls/min/1.73m²) per SD	5678	0.27 (0.15-0.46)	<0.001	0.23 (0.11-0.51)	<0.001
Squared eGFR (mls/min/1.73m²) Per SD	5678	1.09 (1.01-1.15)	0.026	1.16 (1.04-1.29)	0.009

*Age adjusted for sex and sex for age: [#]Fully Adjusted Hazard Ratios adjusted for age, sex, HIV status, hypertension, diabetes mellitus, BMI, marital status, alcohol use and smoking. **SD=22.02 mls/min/1.73m²**

In some covariates, a relevant number of data are missing (marital status, alcohol consumption, smoking status, socioeconomic status, hypertension, and diabetes mellitus). The authors should address how many participants were included in the regression models (age and sex, and fully adjusted), see results, table 2 and abstract. These data are important numbers. The number of 5678 participants (see abstract) might misleading regarding the main purpose of the study. Additionally, these data should also be shown in the study flow, i.e. how many participants were available for the regression models. Further, how was the follow-up on the people who migrated? Were these parts of the analyses?

Thank you very much for these suggestions. We have included the number of participants who were included in the final model in the abstract- **see page 2**. We have also included the numbers and percentages of participants for each of the covariates for both the unadjusted and adjusted models. Please see tables 1 and 2 in the results section for this information on **pages 10 and 12** respectively. 3102 participants were included in the fully adjusted model and these have been highlighted in the manuscript and the abstract on **page 2 and page 12**.

We included participants up to their last point of follow-up (contact). Participants who migrated formed part of the analysis for the time they were present in the area. This is a dynamic cohort and people leave while others come in. Since we did not have new measurements of creatinine, we are unable to provide information on participants of inward migration unless we had their baseline creatinine (that is if they left and came back at another time) their vital status of alive was put into consideration.

If the authors make a statement such as “Comparison of characteristics with included participants showed that those with missing data were younger but otherwise there were no important differences (Supplementary Table 1)”, then p-values should be shown.

Thank you for this observation, Supplementary table 1 has been revised to include a p-values in the table for all variables for a proper comparison. See supplementary table 1 and its legend on **pages 1-2**.

Patients characteristics from the participants analyzed in regression model should be displayed as table

1. In a supplementary table, these data can be compared with people who could not be taken into account due to missing data (with p-values). This table would replace supplementary table 2.

The quality of the figures is rather low. Please revise them. The manuscript should be carefully revised in term of wording and logic within sentences.

Thank you for this suggestion. We have included the numbers and percentages for each of the patients in table 1 and addressed the reviewers concern with more details in table 2. We believe that the total number of the participants enrolled from the study provides a more comprehensive and truer picture of our baseline population from which we drew the participants who made it to the final model. Please see revised tables 1 and 2 on **page 10 and page 12**.

We have included supplementary table one to compare participants in the model and those who were not in the model, see **supplementary table 1**. We have revised the quality of the figures for more clarity and revised the flow of the manuscript into more logical sentences. Please see highlighted changes throughout the manuscript.

Discussion

“However, there are no substantial differences in characteristics between those with complete and incomplete covariate information”. I think, the authors did not compare these two populations. They compared those with complete and those with all complete/non complete information (Supp Table 1).

We adjusted this statement to reflect the findings of the revised supplementary table 1 as shown above.

Others

It seems, trial registration number is missing.

Thank you for this observation, this was however not an interventional trial but a general cohort study and therefore trial registration is not commonly undertaken.

Minor points

Use always the term kidney instead renal (kidney function, kidney impairment and so)

Thank you, this nomenclature changes from renal to kidney has been implemented throughout the manuscript.

Abstract

Paragraph “participants” should be more precise (recruitment between 2011 and 2014), compare background: “We collected data on subsequent mortality for participants who had a baseline creatinine measured in either 2011-2012 or 2014-2015.”

Thank you, the abstract has been corrected to “recruitment between 2011 and 2015” **see page 2** of the abstract under the section of participants.

Please, specify: Overall? Median follow-up (IQ and range): “We registered 140 deaths with a median follow-up of 5.0 years”

This has been corrected to include overall **on page 8**, second paragraph under the results section. We have included an IQR of **5.0 (IQR 3.7-6.0) years**.

Please, add the HR for the subgroup of eGFR ≥ 90 mls/min/1.73m²: “Adjusting for age and sex, HIV, hypertension, diabetes, BMI, marital status, and alcohol and tobacco use participants with eGFR ≤ 45 mls/min/1.73m² had six-fold higher mortality compared to those with eGFR ≥ 90 mls/min/1.73m² (HR 6.12 (95% CI 2.27-16.45)).”

This has been revised to highlight the HR comparing the two subgroups as requested, please see section of results. Please see **page 2** under the section of results in the abstract and **page 10** of results.

“In a prospective cohort with high rates of follow-up we found that baseline kidney function was associated with subsequently increased mortality in a graded manner.” Please revise: baseline kidney function was associated with mortality (because baseline kidney function is not per se associated with increased mortality).

We have revised this sentence to “In a prospective cohort with high rates of follow-up we found that baseline moderate kidney dysfunction was associated with increased mortality.” **See page 2** in the conclusion of the abstract.

Strengths and limitations of this study.

Revise this paragraph. The phrases should be very clear and precise.

We have revised the paragraph on limitations. Please see **pages 13-16**.

Please revise the following paragraph (wording) and please note, in this study, kidney function was not a covariate, but rather the main variable: This is a large well-established population cohort with robust standardized procedures for detailed measurements of covariates such as kidney function and blood pressure, and creatinine was measured according to recommended international standards.

We have revised the sentence on **page 12** to remove kidney function as a covariate.

Background

“Chronic kidney disease (CKD) affects approximately one in every ten adults in high-income countries and is strongly associated with morbidity and mortality 1 2.” Is the study by Stanifer et al the right citation for “high- income countries”?

This has been addressed to reflect the global nature of the statement. “Chronic kidney disease (CKD) affects approximately one in every ten adults across the world and is strongly associated with morbidity and mortality.

“In high-income countries, ESKD is a chronic disease that can be managed with dialysis or kidney transplantation” Is the study by Bukabau et al the right quote for “high-income countries”?

Thank you for pointing this out. We have modified the reference to that by Thurlow JS, et al. Am J Nephrol.

2021, which reflects a more global picture. Please **see page 2** under the section of introduction.

Patient and Public Involvement in Research

“We also undertook a qualitative study to understand the way community members appreciated kidney disease and this work has been published elsewhere”, the part “and this work has been published elsewhere” may be deleted.

Thank you, the extra wording of the sentence has been deleted in the patient and public involvement in research section **on page 5** in the methods section of the paper.

Participants

“who had been recruited from household visits to a homestead in the community”, please revise the wording.

This has been revised to reflect that “participants were recruited from their households to a common homestead called a research hub where the research activities were carried out. Please see additional details on participants.” Please see **page 6** under the section of participants.

Variables

“and classified impaired kidney function in categories analogous to those used to define CKD stages” please cite the more recent guidelines from 2012.

Thank you for noting this. The reference has been updated to site the 2012 KDIGO guidelines, please see section of variables **on page 6**. Below are the details of the reference.

Inker LA, Astor BC, Fox CH, Isakova T, Lash JP, Peralta CA, Kurella Tamura M, Feldman HI. KDOQI US commentary on the **2012 KDIGO** clinical practice guideline for the evaluation and management of CKD. Am J Kidney Dis. 2014 May;63(5):713-35

Statistics

Could you specify what you mean, could you give an example? “Confounders were decided a priori

**based on research regarding associations of chronic conditions with eGFR in this population”
Typing error “s”? 120mls/min/1.73m2**

We have qualified this section by citing additional studies that have established risk factors for kidney diseases such as studies by

1. Kalyesubula R, Hau JP, Asiki G, et al. Impaired renal function in a rural Ugandan population cohort. *Wellcome Open Res* 2018;3:149. doi: 10.12688/wellcomeopenres.14863.3 [published Online First: 2019/06/25]

2. Babua C, Kalyesubula R, Okello E, Cardiovascular **risk factors** among patients with **chronic kidney disease** attending a tertiary hospital in Uganda. *Cardiovasc J Afr.* 2015 Jul-Aug;26(4):177-80. doi: 10.5830/CVJA-2015-045. PMID: 26407219

The sentence now reads:

“Confounders such as diabetes mellitus, hypertension and HIV were included in the analysis model based on previous research showing their association with low eGFR in this population (Babua C et al, 2015 and Kalyesubula R et al, 2018) and after investigation of co-linearity and data sparsity.” Please see **page 7** under the section of statistical analysis.

Results

The incidence rate of death for the participants enrolled in the study was 4 deaths per 1000 person-years at risk (PYAR). Please, specify “overall incidence rate”. For participants with eGFR >90mls/min/1.73m2 the incident rate was about 3 deaths per 1000 (PYAR) person-years at risk and for those <60mls/min/1.73m2 it was 27 deaths per 1000 PYAR. Please avoid imprecise wording in the result section, e.g. “about”.

We have revised this statement to include overall incidence as proposed and removed the word “about” for precision purposes. Please see section under results. Please see **page 8-9**.

Discussion

“which may have led to misclassification of some participants 17.” Replace the reference with the guidelines 2012/13.

Thank you, we have replaced the 2002 guidelines with the 2012/2013 KDIGO guidelines throughout the manuscript. Please see **page 12** under discussion section.

“obesity and diabetes showed low prevalence of impaired renal function and found a protective association with diabetes, possibly due to hyperfiltration.” In this context it seems better not to mention the term “protective”.

Thank you, the word protective has been omitted. Please see **page 13** of the discussion section.

Reviewer: 2

Dr. Anthony Muiro, University of California San Francisco
Comments to the Author:

Multiple studies from the Global North have demonstrated that individuals with Chronic kidney disease (CKD) are at high risk for cardiovascular and all-cause mortality. Even though the burden of CKD in the Global South is rising, few studies have evaluated the long-term outcomes associated with CKD, and none in sub-Saharan Africa. Therefore, the work by Kalyesubula et al, is timely and fills a critical gap in

the **literature.**

Strengths of this study include the use of a well-established large cohort (over 5,000 participants), long-term follow up (with low rates of lost to follow up [other than out migration]), a rural population, and use of standardized kidney function measurements. The use of IDMS calibrated creatinine measurements and the omission of Black race coefficient in CKD-EPI equation to estimate GFR are important additional strengths. The main outcome was ascertained using a validated approach and

seems **appropriate.**

The major weakness of this work is the inability to ascertain causes of death, and lack of albuminuria and follow up kidney function measurements.

Thank you for the review. We agree with the reviewer and intend to critically take this into account (Collection of causes of death, and albuminuria data) in our future work.

We have made extensive changes in all the recommended sections as detailed in responses to each of the issues raised. We have revised the abstract to include the numbers of members included in the final model. We have revised the methods section to include more details on the recruitment process, laboratory and clinical measurements.

We have revised the results section to include analysis on the participants who were included in the final model and also revised the tables to reflect these changes.

We have included a new supplementary table for the extra analysis requested to explain the direct relationship between eGFR and mortality.

Our discussion section has been updated to explain the changes and concerns raised by the reviewers.

I have a few minor questions.

1. Why was eGFR of 45 used as the cutoff point for comparison? why not 60 or 30?

There were few numbers in the eGFR <30mls/min category (7 participants) leading to low power to examine association with mortality. Based on this we decided to use the eGFR <45mls/min category as there was both adequate power and clinical importance.

2. Risk factors: It would be useful to include more details about risk factors control if possible, for example, are markers of HIV control available? Are measures of hypertension control available, this would help strengthen the authors hypothesis "It is possible that repeated screening within this cohort population has led to higher rates of treatment and control than in other populations".

Thank you very much for this suggestion. Unfortunately, we do not have these data available. Yes, it is possible that repeated screening within this cohort population has led to higher rates of treatment and control than in other populations.

3. Out migration: The authors have paid a great deal of attention to the selection bias introduced by the high rate of migration. As they point out, there are important differences between individuals who migrated out of these rural communities and those who remained including age. Is it possible to obtain data from these individuals who have migrated from study communities, for example determine their vital status?

This is a great suggestion. However, it is not possible to get the vital status of those who have migrated since there is no information about the places where the participants might have migrated to. Attempts made to contact them via relatives and neighbors are often futile and would raise another layer of data uncertainty.

4. **On table 1: It would informative to include the total N (%) by eGFR stage**

Thank you, we have modified table 1 in the results section to include the different numbers and percentages by eGFR stage. Please see **page 9** on the results section.

5. It is notable that higher eGFR (>120) was associated with mortality. One possible theory is hyperfiltration (as one would see in patients with sickle cell disease (SCD), though life expectancy for SCD patients is quite low in Uganda and wouldn't expect SCD patients to be included in this study or early diabetes but again the prevalence here is much lower). Another possibility is that low creatinine (i.e. higher eGFR) is indicative of undernutrition, hence the higher mortality.

Thank you very much for these alternative explanations. Sickle cell disease is less common as people get older in Uganda (Ndeezi G, 2016). However, we currently know little about kidney disease and sickle cell disease adults in Uganda. The authors will consider non-creatinine markers of kidney disease such as cystatin C for future exploration.

6. The HR associated with being overweight is notable for being “protective” as is the HR for obese although the CI crosses the null. Would postulate why overweight appears to be protective.

Thank you, the association of eGFR and obesity has been quite controversial but there are some studies that have shown that obesity is associated with hyperfiltration in people of African descent (Wuerzner G, 2010). In our study the hazard ratio estimates of the underweight participants relative to normal participants was 3.59. (HR:3.59, 95% CI:2.20-5.58). This translated into a higher risk of death among the underweight compared to normal weight participants. However, the hazard ratio decreased for higher categories of BMI.

There is clear evidence that obesity (BMI > 30kg/m²) increases incident kidney disease but may be protective in the advanced kidney disease a phenomenon called obesity paradox (Wang Y, 2008; Rhee CM, 2016; Rahimlu M, et al, 2017). In one study BMI had a U-shaped mortality relationship among patients with eGFR above 60mls/min/1.73m² (Lu JL, 2015) with both low (<20kg/m²) and high BMI (>35Kg/m²) having higher mortality across all age groups above 40 years. In Uganda patients with kidney disease often present late with severe uremia, poor appetite and loss of weight. The BMI may therefore be a marker of late presentation and more advanced kidney disease at presentation (Kalyesubula R et al 2016, Namuyimbwa L, et al 2018).

VERSION 2 – REVIEW

REVIEWER	Mayr, Michael University Hospital Basel, Medical Outpatient Clinic
REVIEW RETURNED	15-Nov-2021

GENERAL COMMENTS	To the author The authors performed a prospective general population based cohort study with the aim to analyse the association between baseline kidney function and mortality in a rural region of Uganda. This topic is important as epidemiological and outcome data are still scarce and important for public health strategies. However, in spite of a first revision there are further relevant points that require careful revision. “We included 5678 participants out of the available 5842 participants enrolled from census conducted in 2011-12 and 2014-15”. As I understand there were two cohorts followed-up, one from the census conducted in 2011-12 and one conducted in 2014-15. The authors should describe how many patients were screened and included in cohort 2011-12 and how many in cohort 2014-15. It would be very helpful to show this in figure “study flow” with the numbers (recruitment, enrollment, number of subjects analyzed by the models taking into account missing data). In addition, it is not clear whether the participants from the census 1 and 2 are the same or different, whether there are differences in the area/district
--

	of recruitment or other differences. This should be clarified in the main body of the manuscript, in the study flow and in the abstract. “Our participant response rate at baseline was high, at 97.2%, we included 5678 participants out of the available 5842 participants enrolled from census conducted in 2011-12 and 2014-15.” It is not clear to which census (census 1 or 2) the response rate is related. If the participants from census 1 and 2 are different, the time of follow-up should also be different between these two groups. The authors should comment on this. In the point by point reply the authors state that “We included participants up to their last point of follow-up (contact). Participants who migrated formed part of the analysis for the time they were present in the area. This is a dynamic cohort and people leave while others come in. Since we did not have new measurements of creatinine, we are unable to provide information on participants of inward migration unless we had their baseline creatinine (that is if they left and came back at another time) their vital status of alive was put into consideration.” Please, try to integrate this important message in the methods. I would suggest to omit table 1 and to replace them with supplementary table 1 and to adapt the manuscript and the regression models accordingly, i.e. to perform the analysis considering the 5 eGFR categories. This would be make the manuscript more readable without loss of information. In summary you can omit table 1, table 2 and put suppl table 2 and 4 in the main manuscript as well as the calculation and figure showing a J-shaped association between kidney function and mortality. I don` t understand the methodology behind table 2. How is it possible to adjust for various variables and to keep them as risk factors in the analysis? As I understand it, in model 1 (age and sex adjusted) the model is adjusted for age and sex and therefore sex and age cannot be an “associated”/”risk” factor. The same would apply to the variables for which model 2 (fully adjusted) was adjusted. The incidence rates of death were compared between eGFR >90 ml/min and < 60 ml/min, the risk of mortality between eGFR > 90 ml/min and <45 ml/min. I would suggest to make the comparison between the same eGFR categories. The authors stated that there was a very strong evidence of a non-linear association between eGFR and mortality and added a supplementary table 3 with two regression models. However, the u-shaped relationship is not directly visible in this table. There was a risk reduction in the eGFR model and a risk increase in the squared eGFR. Could the authors explain this observation? Further, the SD should be different between eGFR and squared eGFR? Further, the authors show hazard ratios “age and sex adjusted” and write in a footnote “age adjusted for sex and sex for age”. It is not clear what the authors calculated. Further, the footnote (asterix) has no corresponding asterix in the table. Instead table 3, I would suggest to show the u-shaped relationship
--	--

	in a figure, with the corresponding HR, in the main body of the manuscript. “In an additional analysis, when high eGFR was included as a separate category there was weak evidence of a ‘J-shaped curve’ (Supplementary Table 4).” From the data, I cannot see why there is a weak evidence of a ‘J-shaped curve’? “Because of the U-shaped association between GFR and mortality”/“re-enforcing the U-shaped relationship». This is already an interpretation of the finding of non-linearity and belongs to the discussion and not to the results. In general, the authors should make sure that manuscript version with and without tracked change are identically. In the abstract the authors declare there was a “strong evidence of a linear trend for risk of mortality as kidney function declined (P<0.001)”. In the results the authors stated there was strong evidence (P<0.001) of a non-linear association”. This is confusing. The authors state that comparison of characteristics with included participants showed that those with missing data were younger but otherwise there were no important differences (Supplementary Table 1). In contrast to this statement all p-values in the table were < 0.001. Did the authors include the missing versus non-missing numbers in the calculation? The comparison can only be made without including the missing values (Column n = 2567). The authors use the term “fully adjusted model”. There is no fully adjusted model which is able to adjust for all confounders. I would suggest to use in all tables, figures, text the term “Model 1” and “Model 2” and to declare in footnote (tables) for what variables the model was adjusted. The authors should explain in methods what “Verbal autopsy” means. Concerning the role of the funding source the authors should mention who funded the study. “During follow-up there were 140 deaths with a median follow-up of 5.0 years (IQR 3.7-6.0).” “We registered 140 deaths with a median follow-up of 5.0 (IQR 3.7-6.0) years”. Probably, the authors mean “within a median follow-up...”. Regarding follow-up, also note the comment above (two different time frames of the two census). The information which variables were not available in both census is not consistent. The authors should clarify this: “Information such as smoking, diabetes and blood pressure were not measured in both the 2011 and 2014 surveys of the GPC...”; “We also collected information on smoking, alcohol use, exercise patterns and diet in the census rounds of 2011-12 or 2014-15 (thus this information was not available for all participants for both rounds).”; “Due to missing data for some covariates (such as smoking and alcohol use) which were not collected in both the 2011 and 2014 surveys, we present models adjusted for age and sex, and then additionally for baseline HIV status, hypertension, diabetes mellitus, BMI, smoking, alcohol and marital status.” Further, the wording “Information such as smoking, diabetes and blood pressure were not measured in both the 2011 and 2014 surveys of the GPC so our complete case analysis led to reduction in power for the fully-adjusted model”. This could be misleading and suggesting that in both census the information was not available. The authors should clarify this.
--	--

It is not clear why the missing data are a condition to present adjusted models, confer wording: “Due to missing data for some covariates....., we present models adjusted for age and sex, and then additionally for baseline HIV status, hypertension, diabetes mellitus, BMI, smoking, alcohol and marital status.”

In general, before discussion strengths and limitations, the authors should discuss their findings from the perspective of the existing literature. The authors should discuss why there could be a J-shaped relationship between kidney function and mortality. Further, the authors should carefully revise the wording, and pay attention to a very clear and precise argumentation.

“Our study has a number of strengths. It is large and conducted within a well-established population cohort with robust standardised procedures for detailed measurements of covariates such as kidney function and blood pressure, and creatinine was measured according to recommended standards”. Please, revise the wording.

“We measured kidney function only once while two measures 3 months apart are required to confirm a diagnosis of chronic kidney disease, and we do not have measures of proteinuria, an important early marker of kidney damage (and predictor of mortality), which may have led to misclassification of some participants¹⁹”. I did not find any indications in the manuscript that the baseline eGFR was equated with CKD. Further, there was a stratification according to different eGFR and no CKD classification including albuminuria. So, I don’t see a problem regarding “classification”. However, the authors could say that due to missing values of albuminuria nothing can be said about the association between albuminuria and mortality risk.

“Outward migration by predominantly younger participants leads to a degree of selection bias. It seems, trial registration number is missing...”. “Migration and subsequent loss to follow-up among younger, healthier participants may have led to selection bias with overrepresentation of older participants with health problems including impaired kidney function remaining in the cohort.” The authors should discuss this issue in the same paragraph. Does migration affect the association between eGFR and mortality or only the mortality rate?

“Recent community-based studies of associations of kidney disease suggest important differences in prevalence and associations of impaired kidney function between countries in sub-Saharan Africa. Those including participants from urban areas in South Africa and Nairobi found similar risk factors for those well-established in high-income countries: age, hypertension, diabetes, HIV and female sex ²³. However, cohorts in Malawi and Uganda where the populations are younger, often living in rural areas and with low levels of smoking, obesity and diabetes showed low prevalence of impaired kidney function and found an protective association with diabetes, possibly due to hyperfiltration ^{15 24}.”

The paper of the authors adds nothing to this background.

“The prospective studies.....was not associated with the presence of impaired kidney function or mortality”, Please revise carefully to make your points very clear.

Abstract

“recruited between 2011 and 2015”. I would suggest to describe that recruitment was done in two cohorts 2011/12 and 2014/15. See also the comment above.

In conclusion the authors state “with high rates of follow-up”. However, there are no data in the results in the abstract. In

	general, all points in the paragraph conclusion should be shown in the results Strengths and limitations of this study “Outward migration by predominantly younger participants and use of a single measurement of creatinine without microalbuminuria may have led to a degree of bias”. The authors should explain why this could be a bias in relation to the results found . Table 1: First column n (%) is missing, e.g. 4563 (80.4) Figure legends: “Hazard ratios for the fully adjusted associations of baseline eGFR and mortality in Uganda». Based on the locally collected data, the authors cannot conclude that it is “in Uganda”. Omit “s” mls/min/1.73m2 Acknowledgement Could the authors briefly describe what the role of the cited people was. Regarding Strobe (see also comments above) Abstract: should still be more informative and balanced. Setting and participants: still not fully clear Statistical methods: methodological not clear how the authors handled the variables for which the models were adjusted for. In the manuscript it is not addressed how loss to follow-up was addressed. Participants: numbers of participants in the two census is missing. Further, whether there is an overlap of participants in census 1 and census 2 Descriptive data: missing the follow-up data of participants of census 1 and census 2. Main results: unadjusted estimates are missing Discussion: key results are not well summarized with reference to study objectives Interpretation: Discussion about the generalizability of the results is missing
--	---

REVIEWER	Muiru, Anthony University of California San Francisco
REVIEW RETURNED	02-Nov-2021

GENERAL COMMENTS	N/A
-----

VERSION 2 – AUTHOR RESPONSE

		Point-by-point response
1	The authors performed a prospective general population-based cohort study with the aim to analyse the association between baseline kidney function and mortality in a rural region of Uganda. This topic is important as epidemiological and outcome data are still scarce and important for public health strategies. However, in spite	Thank you for recognising the importance of this data and for providing very detailed feedback to improve it: we have attempted to address all the points you raise as detailed below.

	of a first revision there are further relevant points that require careful revision.	
2	“We included 5678 participants out of the available 5842 participants enrolled from census conducted in 2011-12 and 2014-15”. As I understand there were two cohorts followed-up, one from the census conducted in 2011-12 and one conducted in 2014-15. The authors should describe how many patients were screened and included in cohort 2011-12 and how many in cohort 2014-15. It would be very helpful to show this in figure “study flow” with the numbers (recruitment, enrollment, number of subjects analyzed by the models taking into account missing data). In addition, it is not clear whether the participants from the census 1 and 2 are the same or different, whether there are differences in the area/district of recruitment or other differences. This should be clarified in the main body of the manuscript, in the study flow and in the abstract.	We acknowledge that this was unclear and have attempted to clarify this topic throughout the manuscript. Due to the structure of the data we can access we are not able to disaggregate it by round, and we cannot access the raw data due to COVID-related limits on travelling to the research base and redeployment of staff to work on COVID. As outlined in the cohort description paper there are no substantial differences among adults selected for participation in different medical rounds of the GPC. However, supplementary Table 1 does show differences between people included and not included in the final model: this is because of two factors. Firstly, people without complete data are younger and with better kidney function. This may be partly through chance and because younger people are more likely to be engaged in daily work and not able to stay for complete baseline measures. The second reason is that younger, fitter people are more likely to have out migrated. However, as we discuss in detail below the lack of change of effect estimates between age-sex and confounder adjusted estimates suggests very limited confounding in this study, and therefore the drop in power due to missing data is not a substantial limitation in this work: indeed, that is why we presented both estimates for comparison in the first place. Secondly, while out migration, a problem inherent to all LMIC cohort studies, affects external validity it does not impact the internal validity of this study. While the healthy migrant effect is well known, among the majority of the baseline cohort who have remained at or returned to their home, kidney function is strongly and independently associated with subsequent mortality. We have extended the coverage of these biases in the discussion, with relevant sections summarised in response to related points below.

3	“Our participant response rate at baseline was high, at 97.2%, we included 5678 participants out of the available 5842 participants enrolled from census conducted in 2011-12 and 2014-15.” It is not clear to which census (census 1 or 2) the response rate is related.	Thank you: as above we agree this is not clear and have removed these numbers.
4	If the participants from census 1 and 2 are different, the time of follow-up should also be different between these two groups. The authors should comment on this.	Thank you, we have added this as a limitation to the discussion. The text is highlighted below in relation to your concerns about the impact of missing data.
5	In the point by point reply the authors state that “We included participants up to their last point of follow-up (contact). Participants who migrated formed part of the analysis for the time they were present in the area. This is a dynamic cohort and people leave while others come in. Since we did not have new measurements of creatinine, we are unable to provide information on participants of inward migration unless we had their baseline creatinine (that is if they left and came back at another time) their vital status of alive was put into consideration.” Please, try to integrate this important message in the methods.	Thank you, we have added the key information here to the methods section. Statistical Analysis Page 7: “In Cox regression we included participants up to their last point of follow-up, so participants who later migrated contributed follow-up time to the analysis while they were present in the area. Participants included in the baseline round also contributed follow-up time if they returned to the area after a period of out-migration.”
6	I would suggest to omit table 1 and to replace them with supplementary table 1 and to adapt the manuscript and the regression models accordingly, i.e. to perform the analysis considering the 5 eGFR categories. This would be make the manuscript more readable without loss of information. In summary you can omit table 1, table 2 and put suppl table 2 and 4 in the main manuscript as well as the calculation and figure showing a J-shaped association between kidney function and mortality.	We appreciate this suggestion and have considered it carefully. Since the analysis was pre-planned without a separate high GFR category, with the 5-category analysis as a secondary analysis, we would prefer to keep the structure of the paper as planned. In addition, the 5-category analysis also adds complexity through different interpretations of the age-sex and fully adjusted model where the 95% CI cross 1.
7	I don't understand the methodology behind table 2. How is it possible to adjust for various variables and to keep them as risk factors in the analysis? As I understand it, in model 1 (age and sex adjusted) the model is adjusted for age and sex and therefore sex and age cannot be an “associated”/“risk” factor. The same would	We agree and apologise that this was not clear. In the age-sex adjusted model, the HRs for eGFR categories are adjusted for age and sex: by definition, therefore, the coefficients for age are also adjusted for eGFR and sex, while those for sex are adjusted for eGFR category and age. We agree this may be confusing terminology for non-epidemiological readers so we have removed it. In addition, we have removed the potentially misleading univariable

apply to the variables for which model 2 (fully adjusted) was adjusted.	associations from the age-sex adjusted column in Table 2. The ‘fully adjusted’ column shows the mutually adjusted HRs for each of the covariates included in the model. The key reason for including both these models is to show how little the HRs for the eGFR categories change after additional adjustment for the further potential confounders (such as diabetes) compared to age and sex alone, despite the drop in number of participants included in the model due to missing data. This sequential adjustment approach to show the effect of additional confounders is something we have commonly used before, for example here. We have added to the text for more explanation about this: Statistical analysis, Page 7: “Due to incomplete data for some covariates, we present sequentially adjusted models to demonstrate the effect of additional covariate adjustment on the relationship between eGFR and mortality. Thus, we present models adjusted for age and sex, and then additionally for baseline HIV status, hypertension, diabetes mellitus, BMI, smoking, alcohol and marital status.” Discussion, Page 12 “In addition, participants in the 2011-12 round, who were more likely to have had complete data for potential confounders, also had longer follow-up time over which they could have died. However, marginal differences between the effect estimates for the association of eGFR categories with mortality between the age-sex adjusted and fully adjusted models suggests that neither confounding nor selection bias due to variable follow-up have substantially impacted on our findings of the association between baseline kidney function and mortality”
--	---

8	The incidence rates of death were compared between eGFR >90 ml/min and < 60 ml/min, the risk of mortality between eGFR > 90 ml/min and <45 ml/min. I would suggest to make the comparison between the same eGFR categories	Thank you. There was no reference group for this descriptive data but we did not report all the groups: this has now been amended and rates are reported for each category. Results, Page 8 “The incidence rate for participants with eGFR >90ml/min/1.73m², eGFR between 60-90 ml/min/1.73m², eGFR between 45-60 ml/min/1.73m², and eGFR <45 ml/min/1.73m² was 3, 9, 19, and 63 deaths per 1000 PYAR, respectively.”
9	The authors stated that there was a very strong evidence of an non-linear association between eGFR and mortality and added a supplementary table 3 with two regression models. However, the u-shaped relationship is not directly visible in this table. There was a risk reduction in the eGFR model and a risk increase in the squared eGFR. Could the authors explain this observation? Further, the SD should be different between eGFR and squared eGFR? Further, the authors show hazard ratios “age and sex adjusted” and write in a footnote “age adjusted for sex and sex for age”. It is not clear what the authors calculated.	We have considered this issue carefully. We did not seek to assess the shape of the association between eGFR and mortality: we sought to determine if low eGFR was associated with mortality in this cohort. Very strong data globally suggest that the true relation is ‘J-shaped’. While there is some evidence that this is the case in our study, a very strong association between young age and high eGFR, as well as large measurement error at high eGFR mean that in this cohort we are not powered to assess this, and indeed it was planned as a secondary analysis. In this table we sought to identify the ‘best fit’ slope of the eGFR to mortality in response to previous reviewer’s comments to address the question of the ‘J-shape’ but on reflection believe that additional analysis detracts from the simple and robust findings of our study and have removed it. We will return to this in future follow-up of a larger cohort.
	Further, the footnote (asterix) has no corresponding asterix in the table.	Apologies: the tables have been simplified and made consistent throughout the paper and table footnotes have been checked or removed where not needed, throughout the paper.
10	Instead table 3, I would suggest to show the u-shaped relationship in a figure, with the corresponding HR, in the main body of the manuscript.	Thank you: we have now added an additional graph (supplementary Figure 4), alongside the data. As explained above, as the 5-category analysis was intended as a secondary analysis so we

		prefer to keep the structure of the paper as planned and have tried to focus on the original hypothesis for the data presented in the main body of the paper.
11	“In an additional analysis, when high eGFR was included as a separate category there was weak evidence of a ‘J-shaped curve’ (Supplementary Table 4).” From the data, I cannot see why there is a weak evidence of a ‘J-shaped curve’?	We have now added an additional graph (supplementary Figure 4) to address this and describe it in the text as a ‘reverse J-shaped curve’. We feel that ‘weak evidence’ is an appropriate level of caution between dismissing and overstating our findings, since the lower bound of the 95% CI cross the null in the fully adjusted model and yet we know that this is the case in larger studies from high-income countries.
12	“Because of the U-shaped association between GFR and mortality”/”re-enforcing the U-shaped relationship». This is already an interpretation of the finding of non-linearity and belongs to the discussion and not to the results.	Thank you: as mentioned above we have removed this section from the discussion but now include this text: Discussion, Page 14 “These studies also demonstrate an increase in mortality with people at the highest level of eGFR, in a reverse ‘J-shaped curve’. Our findings are again consistent, demonstrating weak evidence of an increase in mortality for people with the highest category of baseline eGFR. Further confirmation will be needed in larger cohorts with longer follow-up.”
13	In general, the authors should make sure that manuscript version with and without tracked change are identical	Apologies. As this version of the manuscript has been extensively edited we supply a clean version for ease of reading with an additional tracked changes version.
14	In the abstract the authors declare there was a “strong evidence of a linear trend for risk of mortality as kidney function declined (P<0.001)”. In the results the authors stated there was strong evidence (P<0.001) of a non-linear association”. This is confusing.	Thank you: as we have removed the analysis related to the shape of GFR relation to mortality (as per point 9) we now only refer to the test for trend in the abstract and the results and hope this is now clear.
15	The authors use the term “fully adjusted model”. There is no fully adjusted model which is able to adjust for all confounders. I would suggest to use in all tables, figures, text the term “Model 1” and “Model 2” and to declare in footnote (tables) for what variables the model was adjusted.	The fully adjusted model is a complete case analysis that includes the a priori selected confounders. As mentioned above, we have explained the rationale for this approach in more detail in the methods section. This terminology appears to be acceptable to the BMJ group, e.g. here .

16	The authors should explain in methods what "Verbal autopsy" means.	Thank you, we have added this.
17	Concerning the role of the funding source the authors should mention who funded the study.	This is outlined in the funding section at the end of the paper: we are happy to amend it according to editorial guidance.
	"During follow-up there were 140 deaths with a median follow-up of 5.0 years (IQR 3.7-6.0)." "We registered 140 deaths with a median follow-up of 5.0 (IQR 3.7-6.0) years". Probably, the authors mean "within a median follow-up...". Regarding follow-up, also note the comment above (two different time frames of the two census).	Thank you, we have changed the wording to clarify the follow up in the abstract and the results section. As mentioned in point 7 we have added discussion about the issues related to different lengths of follow-up from participants in each round in more depth in the discussion.
18	The information which variables were not available in both census is not consistent. The authors should clarify this: "Information such as smoking, diabetes and blood pressure were not measured in both the 2011 and 2014 surveys of the GPC..."; "We also collected information on smoking, alcohol use, exercise patterns and diet in the census rounds of 2011-12 or 2014-15 (thus this information was not available for all participants for both rounds)."; "Due to missing data for some covariates (such as smoking and alcohol use) which were not collected in both the 2011 and 2014 surveys, we present models adjusted for age and sex, and then additionally for baseline HIV status, hypertension, diabetes mellitus, BMI, smoking, alcohol and marital status." Further, the wording "Information such as smoking, diabetes and blood pressure were not measured in both the 2011 and 2014 surveys of the GPC so our complete case analysis led to reduction in power for the fully-adjusted model". This could be misleading and suggesting that in both census the information was not available. The authors should clarify this.	Thank you: as mentioned above we have aimed to simplify this topic and make the comments about incomplete data consistent throughout the revised manuscript in the text (including with restructured methods) and the tables. Methods, Page 6 "Further to these near complete characteristics, in keeping with the different disease areas investigated in different rounds of the GPC, we also collected information on, marital status, smoking, alcohol use, diabetes and blood pressure. This information was the focus of the 2011-12 round so data collection was near complete but was also collected, though not completely, in 2014-15."
19	It is not clear why the missing data are a condition to present adjusted models, confer wording: "Due to missing data for some covariates....., we present models adjusted for age and sex, and then additionally for baseline HIV status, hypertension, diabetes mellitus, BMI, smoking, alcohol and marital status."	Thank you: as mentioned above we have aimed to clarify the rationale and wording about why we have presented sequentially adjusted models. The discussion is based on the BMJ guidance for authors about the structured discussion as

	In general, before discussion strengths and limitations, the authors should discuss their findings from the perspective of the existing literature.	here which places strengths and limitations second, after statement of principle findings. We are happy to change this according to editorial guidance.
20	The authors should discuss why there could be a J-shaped relationship between kidney function and mortality. Further, the authors should carefully revise the wording, and pay attention to a very clear and precise argumentation	Thank you: we present our revised wording in the response to point 12.
21	“Our study has a number of strengths. It is large and conducted within a well-established population cohort with robust standardised procedures for detailed measurements of covariates such as kidney function and blood pressure, and creatinine was measured according to recommended standards”. Please, revise the wording	Unfortunately, we are not sure what you would like us to revise here and believe this statement to be correct. We are happy to amend it according to editorial guidance.
22	“We measured kidney function only once while two measures 3 months apart are required to confirm a diagnosis of chronic kidney disease, and we do not have measures of proteinuria, an important early marker of kidney damage (and predictor of mortality), which may have led to misclassification of some participants ¹⁹ . I did not find any indications in the manuscript that the baseline eGFR was equated with CKD. Further, there was a stratification according to different eGFR and no CKD classification including albuminuria. So, I don’t see a problem regarding “classification”. However, the authors could say that due to missing values of albuminuria nothing can be said about the association between albuminuria and mortality risk.	The reason for this terminology and mention of lack of repeat measures in the limitations is that in previous publications we have received substantial reviewer criticism for using CKD as terminology instead of eGFR, or for not having measured CKD per se. Therefore, we prefer to keep the terminology as we have written it and are very grateful that Dr Mayr does not see it as a limitation! We have added changed the discussion of the implications of missing proteinuria data: Discussion, Page 13 “In addition, we do not have measures of proteinuria, an important early marker of kidney damage so cannot comment on the association with mortality in this setting.”
23	“Outward migration by predominantly younger participants leads to a degree of selection bias. It seems, trial registration number is missing...”. “Migration and subsequent loss to follow-up among younger, healthier participants may have led to selection bias with overrepresentation of older participants	Thank you: we have simplified and unified the discussion about the potential impact of migration. Discussion, Page 13

	with health problems including impaired kidney function remaining in the cohort.” The authors should discuss this issue in the same paragraph. Does migration affect the association between eGFR and mortality or only the mortality rate?	“Migration and subsequent loss to follow-up among younger, healthier participants may have led to selection bias with overrepresentation of older participants with health problems including impaired kidney function remaining in the cohort. However, people with chronic health problems often return home so we anticipate capturing the majority of deaths among the baseline cohort, even among those who previously migrated.”
24	Recent community-based studies of associations of kidney disease suggest important differences in prevalence and associations of impaired kidney function between countries in sub-Saharan Africa. Those including participants from urban areas in South Africa and Nairobi found similar risk factors for those well-established in high-income countries: age, hypertension, diabetes, HIV and female sex 23. However, cohorts in Malawi and Uganda where the populations are younger, often living in rural areas and with low levels of smoking, obesity and diabetes showed low prevalence of impaired kidney function and found an protective association with diabetes, possibly due to hyperfiltration 15 24.” The paper of the authors adds nothing to this background.	Thank you, we accept that the point we were trying to make was not clear in this paragraph. The fact that these are (collaborative) papers in which we are involved is incidental and we have not deliberately self-cited. We have added a further sentence to clarify this: Discussion Page 13 “These findings suggest that the aetiology of kidney disease in rural sub-Saharan Africa is unclear and highlight the possibility that the mechanism by which kidney function is associated with increased mortality may differ from that in high-income countries.”
25	“ “The prospective studies.....was not associated with the presence of impaired kidney function or mortality”, Please revise carefully to make your points very clear.	Thank you, we have corrected this grammatical error and others in the discussion.
26	The authors stated that there was a very strong evidence of an non-linear association between eGFR and mortality and added a supplementary table 3 with two regression models. However, the u-shaped relationship is not directly visible in this table. There was a risk reduction in the eGFR model and a risk increase in the squared eGFR. Could the authors explain this observation? Further, the SD should be different between eGFR and squared eGFR? Further, the authors show hazard ratios “age and sex adjusted” and write in a footnote “age adjusted for sex and sex for age”. It is not clear what the authors calculated. Further, the footnote (asterix) has no corresponding asterix in the	Thank you: as addressed in response to point 9 we have removed the section regarding modelling the shape of the relationship between eGFR and mortality, and added the figure for the five eGFR category analysis as addressed in response to point 11. We explain why we would prefer to keep this analysis as a secondary analysis in response to point 6.

	table. Instead table 3, I would suggest to show the u-shaped relationship in a figure, with the corresponding HR, in the main body of the manuscript.	
27	Abstract “recruited between 2011 and 2015”. I would suggest to describe that recruitment was done in two cohorts 2011/12 and 2014/15. See also the comment above. In conclusion the authors state “with high rates of follow-up”. However, there are no data in the results in the abstract. In general, all points in the paragraph conclusion should be shown in the results	Thank you: we have now edited the abstract to address these points, to provide more detail on methods and results, and to not draw conclusions not supported by data in the abstract.
28	Strengths and limitations of this study “Outward migration by predominantly younger participants and use of a single measurement of creatinine without microalbuminuria may have led to a degree of bias”. The authors should explain why this could be a bias in relation to the results found .	This section has been rewritten in the requested ‘Summary Box style’. The bias is addressed in the discussion as previously outlined.
29	Table 1: First column n (%) is missing, e.g. 4563 (80.4)	Thank you: we have added these
30	Figure legends: “Hazard ratios for the fully adjusted associations of baseline eGFR and mortality in Uganda». Based on the locally collected data, the authors cannot conclude that it is “in Uganda”.	Thank you: we have amended this and it now reads: Figure 2: Title: Hazard ratios and 95% CI for the fully adjusted associations of baseline eGFR and mortality in a rural Ugandan population cohort
31	Omit “s” mls/min/1.73m ²	We have edited these
32	Acknowledgement Could the authors briefly describe what the role of the cited people was.	Thank you: we have now added this to the manuscript.
33	Regarding Strobe (see also comments above) Abstract: should still be more informative and balanced. Setting and participants: still not fully clear Statistical methods: methodological not clear how the authors handled the variables for which the models were adjusted for. In the manuscript it is not addressed how loss to follow-up was addressed. Participants: numbers of participants in the	Thank you: in this carefully revised version we believe we have now addressed all these points. We do not wish to present unadjusted estimates of the association between eGFR and mortality since age and sex are such strong confounders of both exposure and outcome: crude results would not be meaningful.

	twocensus is missing. Further, whether there is an overlap of participants in census 1 and census 2 Descriptive data: missing the follow-up data of participants of census 1 and census 2. Main results: unadjusted estimates are missing Discussion: key results are not well summarized with reference to study objectives Interpretation: Discussion about the generalizability of the results is missing	
--	---	--